# Rp3: Ribosome profiling-assisted proteogenomics improves coverage and confidence during microprotein discovery

Eduardo Vieira de Souza [1,2,3], Angie L. Bookout[4], Christopher A. Barnes [4], Brendan Miller[3], Pablo Machado[1,2], Luiz A. Basso[1,2], Cristiano V. Bizarro [1,2] ✉ & Alan Saghatelian [3] ✉

There has been a dramatic increase in the identification of non-canonical translation and a significant expansion of the protein-coding genome. Among the strategies used to identify unannotated small Open Reading Frames (smORFs) that encode microproteins, Ribosome profiling (Ribo-Seq) is the gold standard for the annotation of novel coding sequences by reporting on smORF translation. In Ribo-Seq, ribosome-protected footprints (RPFs) that map to multiple genomic sites are removed since they cannot be unambiguously assigned to a specific genomic location. Furthermore, RPFs necessarily result in short (25-34 nucleotides) reads, increasing the chance of multi-mapping alignments, such that smORFs residing in these regions cannot be identified by Ribo-Seq. Moreover, it has been challenging to identify protein evidence for Ribo-Seq. To solve this, we developed Rp3, a pipeline that integrates proteogenomics and Ribosome profiling to provide unambiguous evidence for a subset of microproteins missed by current Ribo-Seq pipelines. Here, we show that Rp3 maximizes proteomics detection and confidence of microprotein-encoding smORFs.

Small open reading frames (smORFs, <300–450 nucleotides)[1–4] were excluded from typical genome annotation workflows because there were no methods to validate their translation, which led to the concern that most smORFs were false positives[5,6]. The advent of Ribosome profiling, or Ribo-Seq, provided empirical data to identify actively translated smORFs and provide an accurate snapshot of translation at sub-codon resolution[7,8]. The ability to accurately define translated smORFs has led to greater and greater efforts in this area to identify and functionally characterize these genes.

The assignment of smORFs occurs after Ribo-Seq data is fed into a bioinformatics workflow. A variety of different scoring algorithms are used to identify smORFs, such as RibORF[9], Ribocode[10], RiboDIPA[11],

ORFRater[12], RiboTaper[13], ORFscore[14], RP-BP[15], and PRICE[16]. These tools differ substantially in their approach, but most tools were designed for the de novo annotation of the translatome by assessing the 3-nucleotide periodicity, although machine learning algorithms trained on available annotation data are also common. The performance of these tools varies in many metrics, and ORFRater performance is also heavily affected by the usage of the drug harringtonine during sample preparation, which is a protein synthesis inhibitor that stalls the ribosome on the initiation codon to better identify the start of a (sm)ORF[12].

Although many pipelines are being successfully used for the identification of actively translated smORFs, there are some

[1]Centro de Pesquisas em Biologia Molecular e Funcional (CPBMF) and Instituto Nacional de Ciência e Tecnologia em Tuberculose (INCT-TB), Pontifícia Universidade Católica do Rio Grande do Sul (PUCRS), Porto Alegre, Brazil. [2]Programa de Pós-Graduação em Biologia Celular e Molecular, Pontifícia Universidade Católica do Rio Grande do Sul, 90616-900 Porto Alegre, Rio Grande do Sul, Brazil. [3]Clayton Foundation Laboratories for Peptide Biology, Salk Institute for Biological Studies, La Jolla, CA, USA. [4]Novo Nordisk Research Center Seattle Inc., Seattle, WA, USA. ✉e-mail: cristiano.bizarro@pucrs.br; asaghatelian@salk.edu

shortcomings that limit smORF annotation. A common problem is the existence of reads that map to multiple locations in the genome during the alignment step of these workflows[17]. These duplicated regions might result from a variety of events, such as whole genome duplication[18,19], recombination[20], retro-transposition[21], and gene duplication[22]. Multi-mapping reads must be addressed with caution during genomics and transcriptomics analysis, as they can confound gene quantification and make genome annotation more difficult.

In a typical RNA-Seq analysis with short-reads, multi-mapping reads make up from 5% to 40% of total mapped reads[17]. This proportion is expected to be bigger in a Ribo-Seq analysis, as reads from ribosome footprints range from 25 to 34 bp[2,7], which increases the chance of them mapping to multiple sites when compared to traditional Illumina short-reads, which are 75–300 bp long[23]. Most of the time, multi-mapped reads are discarded during read assignments in RNA-Seq analyses[17]. In our analysis, we show that the proportion of Ribo-Seq reads that multi-map is much higher than that of RNA-Seq reads, which should complicate ORF discovery when relying solely on translational evidence.

Proteogenomics is a multi-omics approach that integrates genomics, transcriptomics, and proteomics, and can be used for a multitude of tasks, including supporting genome annotation and allowing the identification of microproteins encoded by smORFs[24,25]. The most robust evidence for the translation of a smORF is the direct detection of the resultant microprotein. Therefore, while the total number of smORFs revealed by proteogenomics is eclipsed by the higher number of Ribo-Seq annotated smORFs, there is added value in validating the microprotein translation with proteomics as it provides direct protein evidence.

We reasoned that the integration of proteogenomics alongside Ribo-Seq could provide a solution to overcome the challenge incurred by ambiguous and multi-mapping reads during smORF identification during Ribo-Seq-only analysis. We refer to this as the Ribosome Profiling and Proteogenomics Pipeline (Rp3). Although there are strategies to account for multi-mapping reads, such as ignoring them, splitting the reads across the mapped genes, or using statistical modeling of mapping uncertainty[17], all of which are valid when working with sequencing reads alone, the integration of proteogenomics with Ribo-Seq is an advance since it provides additional layers of evidence by combining protein and translation information to achieve higher confidence in the detection of microproteins.

In this study, we employ the Rp3 workflow and reanalyze previously published datasets to provide additional evidence of smORFs that exist in regions inaccessible by Ribo-Seq alone (Fig. 1). Using the data from these results, we explore how Ribo-seq identifications differ from mass spectrometry peptide evidence of unannotated translated microproteins, including differences in genome and protein sequence composition, transcript isoforms, gene paralogy, and the presence of repeat regions, all of which can preclude the identification of translational events. The data show that Rp3 identifies smORFs and resultant microproteins that are disregarded in most pipelines that rely solely on Ribo-Seq evidence, significantly increasing the number of Ribo-Seq detected smORFs with proteomics detection—the highest confidence set of microproteins.

## Results

### Proteogenomics approach identifies microproteins that could not be found with Ribo-Seq

We have previously used Ribo-Seq to identify unannotated translated smORFs from thermogenic brown adipose tissue (BAT), energy storage-focused white adipose tissue (WAT), and beige adipose tissue[26], which yielded a total of 3877 unannotated sequences, including a bioactive secreted microprotein encoded by the mouse gene Gm8773. In the study, the 3877 unannotated smORFs were translated and appended to the mouse UniProt database to create a custom proteomics database that is able to detect stable microproteins from these sequences. While these results validated the translation of 85 smORFs into microproteins, there is a vast difference in the numbers of microproteins obtained from Ribo-Seq and proteomics. Significantly higher numbers of smORFs from Ribo-Seq are common in the field, and typically only a handful of microproteins are detected using proteomics workflows that incorporate Ribo-Seq smORFs. The lack of detectable microproteins from Ribo-Seq annotated smORFs has been attributed to several factors, including low abundance of microproteins, their length, which results in the generation of few tryptic peptides, and lack of unique tryptic peptides. Indeed, we have detected microprotein via Western blot with antibodies that we have never obtained proteomics data from to further highlight some of the challenges of microprotein detection[27].

Nevertheless, Ribo-Seq does not readily capture all ORFs. Alternative ORFs (AltORFs) overlap but are read in a different reading frame from the canonical ORF, and are easy to detect proteomically but difficult to detect by Ribo-Seq because of two different but overlapping reading frames that come from the same RNA. Several AltORFs have been shown to be functional[5], highlighting the value of identifying these proteins. We also avoided the annotation of upstream overlapping smORFs (uoORFs) in the 3877 smORFs due to similar issues with overlapping transcripts, though uoORFs have been annotated in other RiboSeq studies[1,28]. To determine if we were missing any translated smORFs that were excluded from the Ribo-Seq dataset, we reanalyzed the proteome and secretome data using a proteogenomics approach (Fig. 1).

Proteogenomics relies on custom proteomics databases that are derived from the 3-frame translation of RNA-Seq data, which, in theory, should capture all translated smORFs from the translatome. After excluding peptides from the proteogenomics analysis that matched annotated proteins and keeping only the ones that matched predicted unannotated proteins exclusively, we identified 130 unannotated microproteins (Fig. 2a and Supplementary Data 1). Out of these, 46 had at least one peptide matching proteins in the unreviewed Uniprot database (Supplementary Fig. 1). 33 of these peptides matched sequences with very low annotation scores (1–2), representing protein evidence for poorly characterized proteins. We also ran PRICE, Ribocode (Supplementary Data 1), and RibORF to identify smORFs based solely on RNA-Seq and Ribo-Seq evidence (RS microproteins). Although the number of Ribo-Seq identifications is much higher overall, the microproteins detected by proteomics are valuable since they are validated as stable members of the proteome. Overall, the overlap of the results coming from different pipelines is considerably small, which suggests the approach of Rp3 is orthogonal and complementary to the ones that use Ribo-Seq evidence alone to score smORFs. The proteomics coverage for Ribo-Seq smORFs identified by any of the three RS pipelines is significantly lower than those coming from Rp3 (Fig. 2b). To help understand this, we investigated the classification of smORFs most likely to be found using the different tools (Fig. 2c). The vast majority of smORFs identified with Rp3 reside in genome regions that are either non-coding or that are pseudogenes or retrotransposons (rtORFs). RibORF, Ribocode, and to a lesser extent, PRICE, were more likely to identify upstream ORFs (uORFs). The sequence length for microproteins identified with different pipelines also differ, where Rp3 tends to identify longer microproteins, and most smORFs identified by the other methods tend to be shorter (Fig. 2d).

### Differences in sequence composition may confound mass spectrometry characterization of microproteins from Ribo-Seq results

Unannotated smORFs differ in their amino acid sequence composition when compared to already annotated proteins[2]. We wanted to see if this sequence bias extended to groups of microproteins coming from different approaches, such as proteogenomics and Ribo-Seq pipelines.

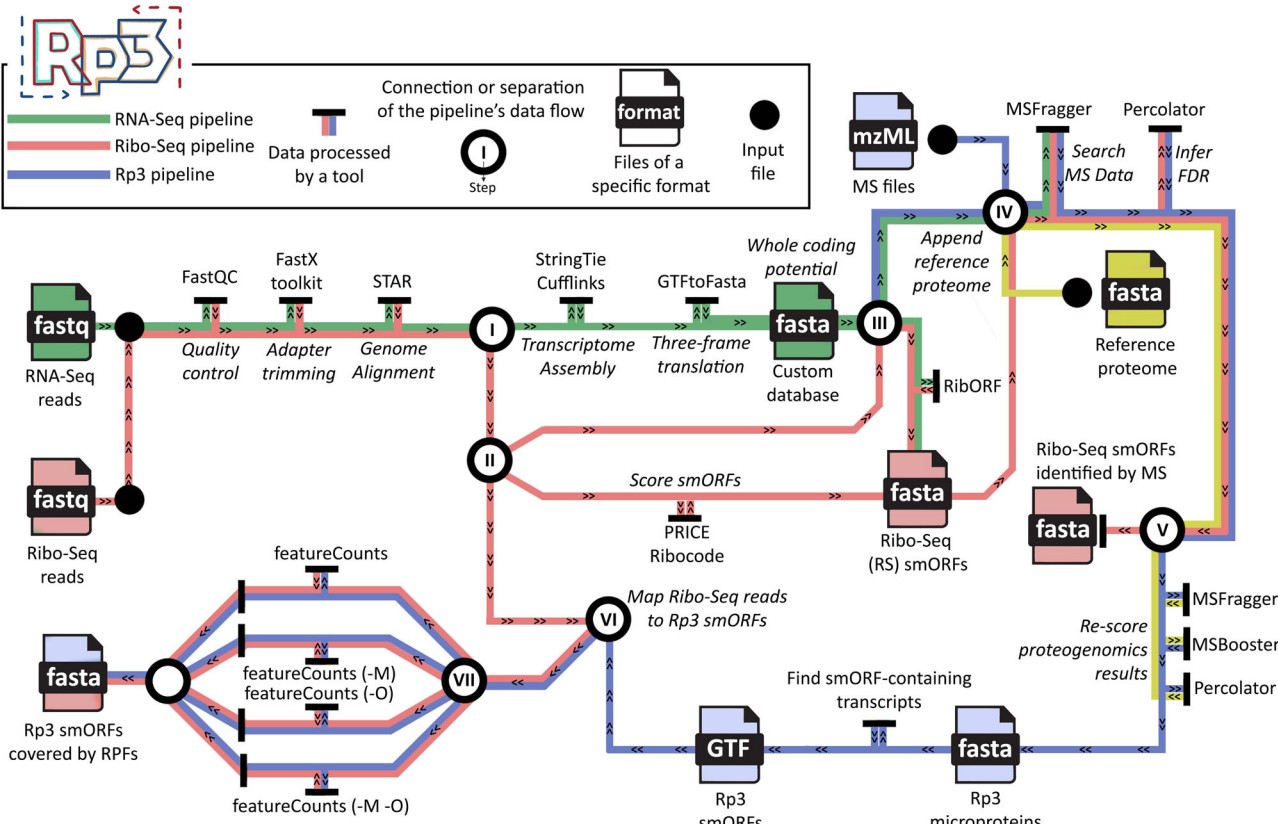

**Fig. 1 | Workflow to reanalyze and overlay datasets.** At the start of the workflow, RNA-Seq and Ribo-seq read undergo the same initial steps, starting with quality control and adapter trimming, followed by the alignment to the genome with STAR[50]. Subsequently, (I), the aligned RNA-Seq reads are used to assemble the transcriptome with StringTie[51], and this assembly is translated to the three-reading frames to predict the whole coding potential of that transcriptome, resulting in the three-frame translated (3FT) database (II) The aligned Ribo-seq reads are then used four times in the pipeline. First, they are used to score the ORFs from the transcriptome with both PRICE and Ribocode. Then, they are used as input for RibORF in step III. The reads are used one last time in step VI. (III) The alignments are scored against the 3FT database using RibORF, resulting in a fasta file containing Ribo-seq smORFs. A reference proteome is then appended to this fasta file to generate a custom database to check the mass spectrometry (MS) coverage for the Ribo-Seq smORFs. Similarly, the reference proteome is appended to the three-frame translated database, now without the Ribo-Seq smORFs, but with every predicted

protein, which is the start of the proteogenomics pipeline. (IV) mzML files containing fragmentation spectra from MS experiments are searched against both databases using MSFragger[40], whose results are filtered with Percolator[41] to obtain an FDR of 1%. This yields two subsets of results (V), Ribo-Seq smORFs covered by both Ribo-seq and MS evidence, and proteogenomics-derived smORFs (Rp3 smORFs), covered by MS evidence alone. The results from the first search are appended to the reference proteome and searched again with MSFragger, now with a reduced database instead of the 3FT. The search results are used as input for MSBooster[42] to predict retention times and then fed to Percolator to assess the FDR. The Rp3 smORF-containing transcripts are located and (VI) the Ribo-seq reads containing secondary alignments are mapped to them using featureCounts[31] (VII): once with default settings, to obtain Rp3 smORFs covered by Ribo-seq reads, and then allowing ambiguous and multi-mapping reads to be included during read counting, resulting in another subset of Rp3 smORFs covered by ambiguous and/or multi-mapped reads.

Indeed, the distribution of amino acids varies among these groups (Fig. 2e), and lysine, which is of utmost importance for bottom-up proteomics[29], is less common in the RS microproteins than it is for the ones found with Rp3, which could contribute to the difficulty in their identification by mass spectrometry. Proline, which is substantially enriched in Ribo-seq microproteins, is known to confound the interpretation of fragmentation spectra. For instance, if the residue is present at the C-terminal, it fragments almost exclusively by cleavage of the amide bond that is adjacent to that proline, resulting mostly in y1 fragment ions[30].

Such amino acid biases could explain the low proteomics coverage for Ribo-seq microproteins, due to an increased difficulty in identifying microproteins rich in proline residues or those with very few tryptic sites. To confirm the latter, we inspected the tryptic peptides (TPs) obtained after an in silico digestion of annotated Rp3 and RS microproteins. The average number of TPs is larger for Rp3 proteins, which makes them more likely to be found in a mass spectrometry experiment (Fig. 2f). The number of TPs for Rp3 microproteins is also higher than those coming from Ribo-seq, which could contribute to their higher proteomics coverage. This could be explained in part by

the tendency of Rp3 to pick up longer microproteins (Fig. 2d), which should produce more TPs after protein digestion. Moreover, the molecular weight for peptides from all three microproteins groups also differs significantly (Fig. 2g). The isoelectric point (pI) is also important for protein identification using mass spectrometry-based proteomics[29], and is significantly lower than the TPs from RS microproteins coming from the RibORF pipeline compared to ones from Rp3 (Fig. 2h).

### Most Ribo-Seq reads cannot be uniquely assigned to smORF-containing transcripts that encode the Proteogenomics-derived (Rp3) microproteins

To understand the minimal overlap between the proteogenomics and Ribo-Seq results, we checked for evidence of translation for the proteogenomics hits. A priori, we expected to find evidence for the translation of Rp3 microproteins but figured they were absent from the RS-microproteins because they fell below a scoring threshold. We mapped Ribo-Seq reads for the Rp3 microproteins back to the transcripts that encode the smORFs. Surprisingly, we measured no Ribo-Seq coverage for the vast majority of proteogenomics smORFs when

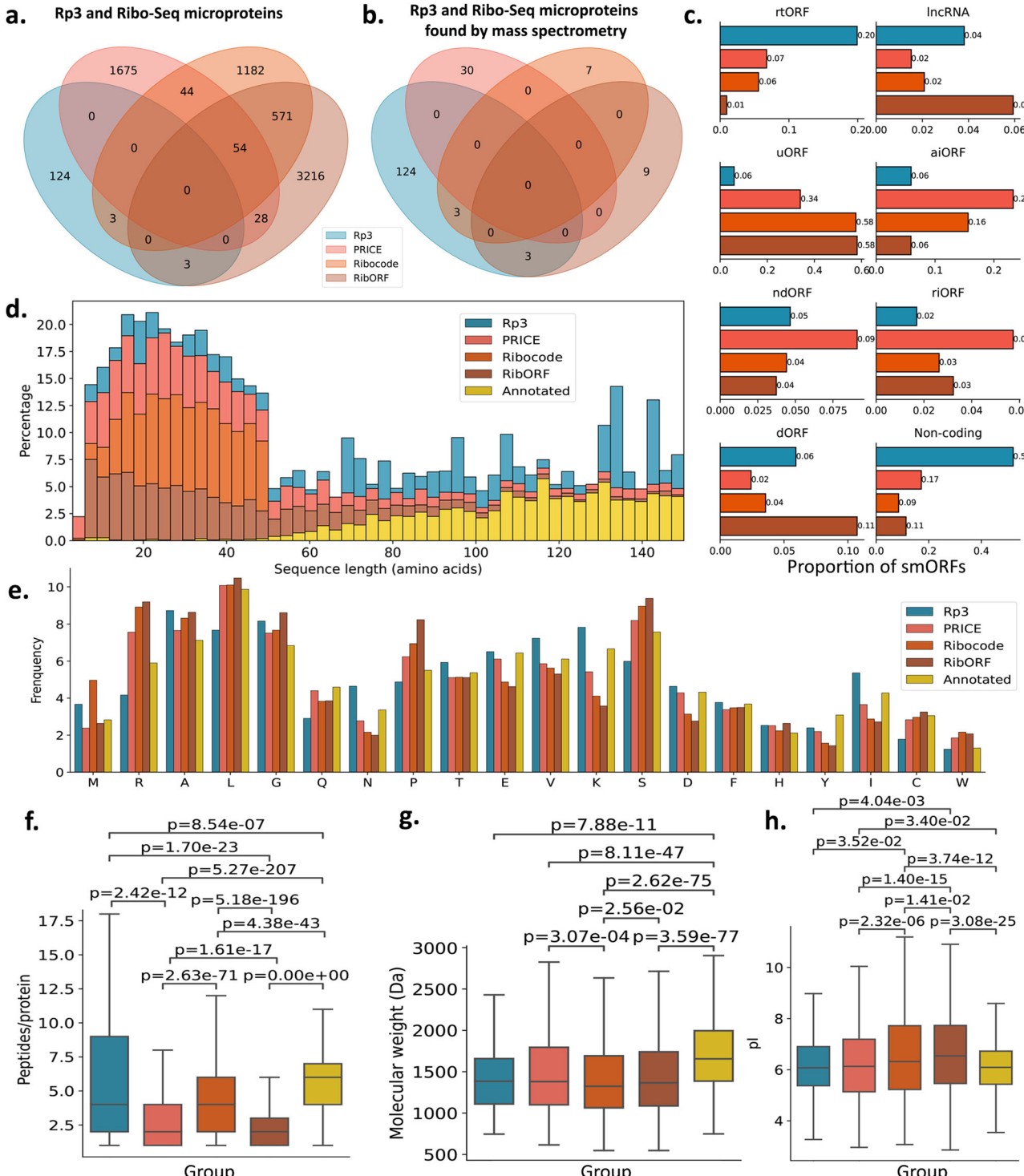

**Fig. 2 | Comparison of microproteins identified by proteogenomics and Ribo-seq. a** Venn diagram showing the intersection of unannotated microproteins identified by Rp3 and PRICE, Ribocode, and RibORF. **b** Intersection between Rp3 microproteins and microproteins from the three Ribo-seq methods that were confirmed by mass spectrometry. **c** Annotation of smORFs types for each tool based on the features they overlap in annotated transcripts, normalized by the number of smORFs found with that specific tool. Non-coding refers to smORFs in regions previously characterized as intergenic regions, within unannotated transcripts. Full definition of smORF types is available in Supplementary Data 2. **d** Sequence length distribution for microproteins identified with the four different tools plus microproteins annotated in SwissProt. **e** Amino acid distribution for each microprotein group. **f** Number of tryptic peptides resulting from the in silico digestion of each microprotein using trypsin as the enzyme for each microprotein group (Kruskal–Wallis, $p < 1.00 \times 10^{-256}$. Dunn's post-hoc exact $p$-values are shown in the plot). **g** Molecular weight for all microprotein groups (Kruskal–Wallis, $p = 1.62 \times 10^{-92}$. Dunn's post-hoc exact $p$-values are shown in the plot). **h** Isoelectric points (pI) for each peptide resulting from the in silico digestion of each microprotein group (Kruskal–Wallis, $p = 1.07 \times 10^{-30}$. Dunn's post-hoc exact $p$-values are shown in the plot). For all box plots, the center line corresponds to the median; box limits represent the upper and lower quartiles; whiskers correspond to 1.5× inter-quartile range. $n = 2$ biological replicates for Ribo-Seq experiments. $n = 5$ biological replicates for mass spectrometry experiments. Results from BAT, beige and WAT were grouped together. Sample size for microproteins compared in **f, g**, and **h** refer to the total number of microproteins in each group, available in Supplementary Data 1, and described in the Venn diagram (**a**). Source data are provided as a Source Data file.

running FeatureCounts[31] with the default settings (Fig. 3b), which should raise concerns about whether the peptide is a true identification or not, due to the lack of translational evidence, despite the presence of proteomics evidence. When we allowed the tool to include ambiguous and multi-mapping reads during read counting, however, we found translational evidence for many Rp3 smORFs in the Ribo-Seq data.

We grouped Rp3 smORFs together based on their detectability by Ribo-Seq. Some smORFs only have Ribo-Seq coverage when allowing for specific parameters during read counting, such as multi-mapping (Fig. 3a, middle), and ambiguous mapping (Fig. 3a, right) which means that these features preclude the identification of translational evidence for smORFs. Most smORFs do not have Ribo-Seq coverage when using default settings during read counting (Fig. 3b, c). We found a significant increase in the RPKMs for RPFs mapping to Rp3 smORFs when allowing multi-mapping (MM), ambiguous mapping (Amb), or both (MM_Amb), and divided them into their respective mapping groups. smORFs and their respective mapping groups are shown in Supplementary Data 1. In these datasets, the overall mapping landscape of Ribo-Seq reads differs drastically compared to RNA-Seq reads (Fig. 3d). As Ribo-Seq reads are much shorter, more than 80% are unassigned due to multi-mapping. This number is considerably lower for RNA-Seq reads, with a distribution ranging from 10% to 30% (t-test, $P < 0.001$). In fact, ~90% of Ribo-Seq reads could not be assigned when running featureCounts with default parameters. For RNA-seq, most reads could not be assigned due to overlapping regions. This could be due to the longer size of transcripts that the RNA-Seq reads are mapping to, compared to the ORFs the Ribo-Seq reads map to. As such, the transcripts should be more likely to overlap with another feature in the genome, requiring the tool to include overlapping features. Out of the multi-mappers from MM and MM_Amb that require multi-mapping specifically to be detected, and thus are absent from the Amb group, only 3 could be identified by Ribocode or RibORF (Fig. 3e). None of these were found with PRICE. The distribution of raw counts covering the Rp3 smORFs resembles a normal distribution (Fig. 3f), while most smORFs have lower RPKM coverage (Fig. 3g).

We found that smORFs in both the Amb and MM_Amb groups are present in regions with a high number of transcript isoforms and/or other overlapping features (Fig. 4a), making it harder for the counting software to distinguish which transcripts the reads might be mapping to. Additionally, MM and MM_Amb groups present a higher percentage of smORFs in repeat regions when compared to Amb and Default groups (Fig. 4b). When not accounting for these features, the counting tool tends to disregard these reads, leading to an apparent lack of translational evidence. This data shows that Rp3 smORFs, for which we have reliable protein evidence, would be disregarded by Ribo-Seq alone, due to limitations in the mapping step. By checking the coverage for Rp3 smORFs while allowing for different mapping characteristics, we are able to detect translational evidence for these sequences as well. Those are backed by the strongest evidence, as we have both proteomics and Ribo-Seq coverage.

To better understand the genomic regions where these smORFs reside based on their mapping classification (Supplementary Fig. 2), we checked the annotated regions where the smORFs overlap in their transcripts (Fig. 4c). Most smORFs in non-coding regions were found to have no Ribo-Seq coverage. The majority of smORFs with Ribo-Seq coverage were rtORFs. smORFs found in an alternative initiation site of another ORF (aiORF) were common among Amb and MM_Amb smORFs, which is expected as they are supposed to be overlapping other genes. Rp3 identified very few smORFs in transcripts annotated as long non-coding RNAs (lncRNAs) and smORFs in transcripts with retained introns (riORFs). The majority of identified uORFs required Amb or MM_Amb parameters to obtain Ribo-Seq coverage. Lastly, we checked the conservation levels for smORFs with no Ribo-Seq coverage and smORFs with coverage from the Rp3 pipeline (Supplementary

Data 1). Overall, a higher proportion of Rp3 smORFs are conserved across different eukaryotes (Fig. 4d), even in distant ones like *Danio rerio* and *Drosophilha melanogaster*, while the ones with no coverage have very few homologs in distant organisms. This suggests that the coverage inferred by our Rp3 pipeline can identify microproteins that are evolutionarily conserved and, thus, more likely to be real.

## Mass spectrometry peptides allow the identification of smORFs with uncertain Ribo-Seq mapping

Using Ribo-Seq evidence alone would result in microproteins being missing (Fig. 3b and c). By contrast, Rp3 combines the strengths of Ribo-Seq and proteogenomics to identify a unique subset of smORFs with a very high level of confidence. As an example of the mapping landscape of short Ribo-Seq reads for Rp3 smORFs, we plotted the position of each smORF identified with both proteogenomics and Ribosome profiling, along with the annotated genes using a circos plot (Fig. 5a). For instance, a smORF found on the reverse strand of chr8, (coordinates 107262838–107263248) would have been filtered out because of multi-mapping (Fig. 5b). The reads for this smORF were mapped to multiple different regions in the genome. None of the Ribo-Seq pipelines were able to identify this smORF due to the high degree of homology to other sequences in the genome. By aligning the microprotein peptide sequence identified from a high-quality MS/MS spectrum (Fig. 5c) across 50 genomic locations with high homology to this protein (full multiple sequence alignment available in Supplementary Fig. 3), it shows that the only nucleic acid sequence that is able to produce this particular microprotein is found at -chr8:107262838–107263248 (Fig. 5d). This example highlights the potential of Rp3 to reveal microproteins with translational and proteomics evidence—the highest confidence group of smORFs/microproteins. It exemplifies the challenges imposed by working solely with Ribo-Seq reads and how proteogenomics can circumvent these limitations.

## Proteogenomics allows the identification of a unique subset of peptides presented in the HLA complex

Peptidomics studies from human leukocyte antigen (HLA) complex provide a rich source of potentially immunogenic microproteins and were used before to characterize smORFs identified by Ribo-Seq. Here, we reanalyzed published datasets[2,32] to determine whether using Rp3 would result in the identification of additional microproteins that were missed using the Ribo-Seq-only pipeline (Fig. 6a). After translating the previously assembled transcriptomes for HeLa-S3, K562 and HEK293T cell lines to the three reading frames, we performed a proteogenomics search with Rp3. Additionally, we used PRICE, Ribocode, and RibORF to score smORFs in these assemblies using RNA-Seq and Ribo-Seq evidence alone (Supplementary Data 2). After running Rp3, we identified 692 unannotated Rp3 microproteins with strong affinity to HLA Class I (Supplementary Data 2), 150 of which were shared with results from at least one of the Ribo-Seq pipelines (Fig. 6b). This overlap was maintained after matching fragmentation spectra from HLA peptidomics experiments with the microproteins coming from the Ribo-Seq pipelines, but the overall proportion of RS microproteins covered by mass spectrometry evidence was still low, following the same patterns of the mouse datasets (Fig. 6c). After checking for Ribo-Seq coverage using the same approach as in Fig. 3b, we found a similar pattern regarding mapping limitations (Fig. 6d), with hundreds of Rp3 smORFs present in regions affected by ambiguous and/or multi-mapping (Fig. 6e). Out of 31 multi-mapping smORFs identified with Rp3, which were found in the MM and/or MM_Amb groups but not Amb, only a few could be identified by the Ribo-Seq pipelines, with RibORF identifying the highest number (Fig. 6f). The distribution of raw counts follows a normal distribution (Fig. 6g), whereas most smORFs have a tendency for lower RPKM values (Fig. 6i). In concordance to the distribution of RPKM values for the mouse datasets (Fig. 3g), the distribution of

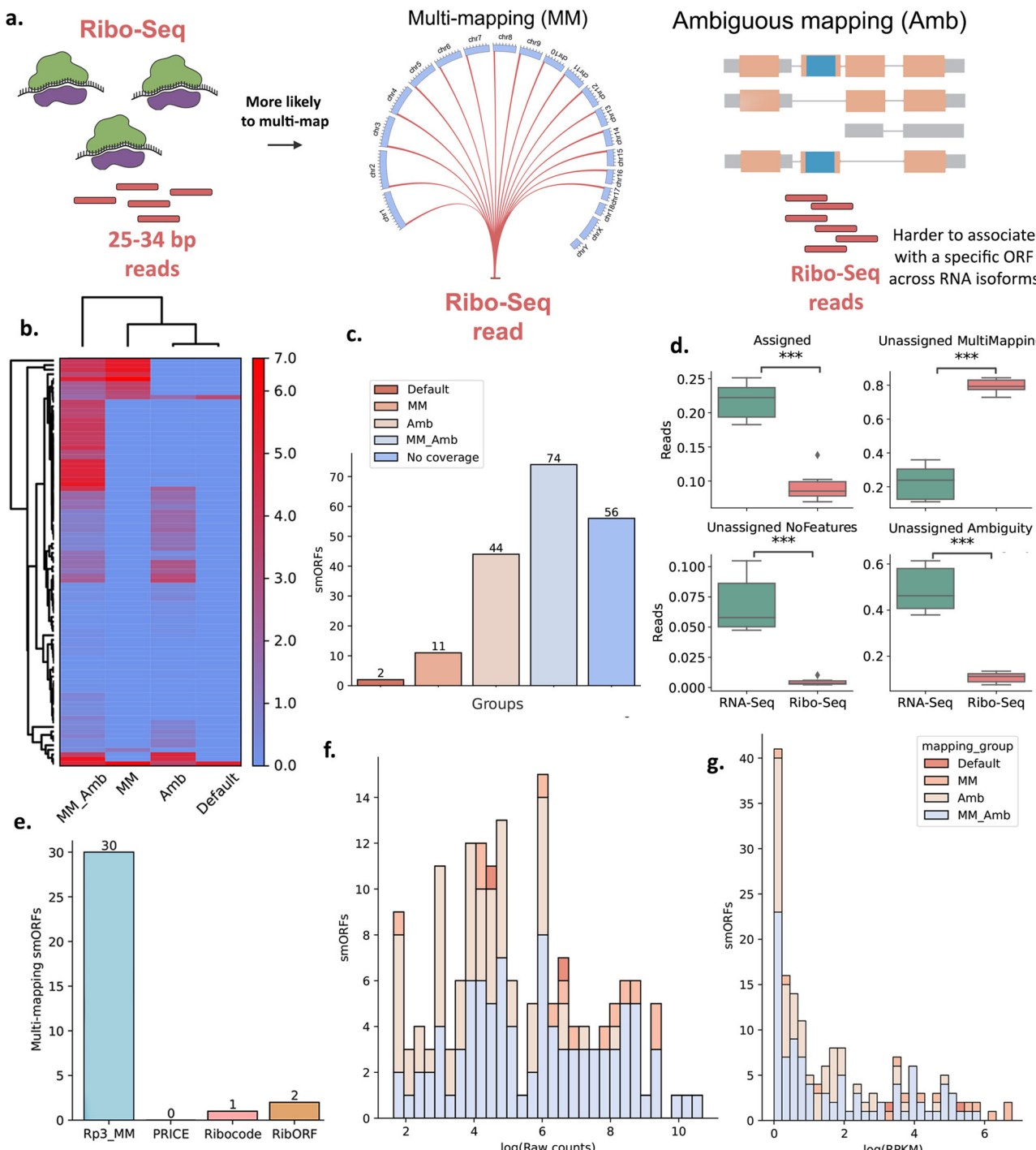

**Fig. 3 | Ribo-Seq coverage for Rp3 smORFs. a** Schematic demonstrating the challenges imposed by short read lengths of Ribosome Protected Fragments (RPF) regarding ambiguous and multi-mapping. **b** Heatmap showing the Ribo-Seq coverage (log RPKM) for Rp3 smORFs. Groups are classified based on the parameters allowed during read counting. Amb: featureCounts ran with the ambiguous mapping parameter set to True (-O); Default: featureCounts executed with default settings; MM: multi-mapping allowed (-M); MM_Amb: both ambiguous and multi-mapping allowed (-O -M) during read counting. smORFs that were found using one of these sets of parameters were grouped together for subsequent analyses. **c** Absolute number of smORFs identified for each mapping group. **d** Number of unassigned reads for Ribo-Seq and RNA-Seq experiments due to multi-mapping and ambiguous mapping when running featureCounts with

default parameters. $T$-test, $p^{***} = 1.14 \times 10^{-08}$ (unassigned ambiguity), $p^{***} = 3.77 \times 10^{-9}$ (Assigned), $p^{***} = 4.08 \times 10^{-11}$ (unassigned multi-mapping), $p^{***} = 1.01 \times 10^{-06}$ (unassigned no features). $n = 2$ biological replicates for BAT, beige, and WAT mouse adipose tissues for both RNA-Seq and Ribo-Seq experiments. The three conditions were grouped together when comparing the sequencing techniques. **e** Which of the Rp3 multi-mapping smORFs, including those in MM or MM_Amb groups but absent from Amb, that were also identified using Ribo-Seq-only pipelines. Distribution of raw read counts (**f**) and RPKM values (**g**) for Rp3 smORFs. **g** For all box plots, the center line corresponds to the median; box limits represent the upper and lower quartiles; whiskers correspond to the 1.5× interquartile range. Source data are provided as a Source Data file.

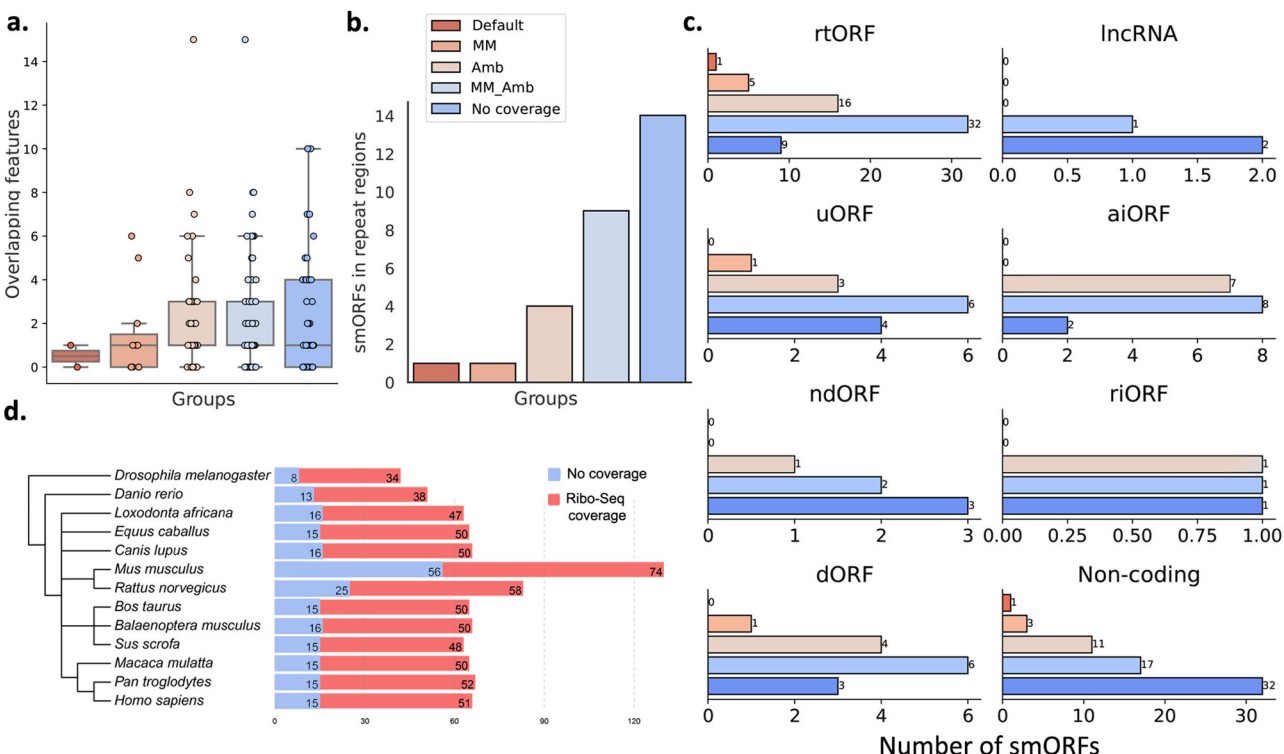

**Fig. 4 | Genomic landscape of Rp3 smORFs based on their mapping classification. a** Distribution of overlapping features for smORFs based on their genome location. $n = 2$ biological replicates for RNA-Seq and Ribo-Seq experiments, and $n = 5$ for mass spectrometry-based proteomics experiments, for each of the three adipose tissue phenotypes. All microproteins identified with Rp3 from any adipose tissue phenotype were combined to perform the comparisons. Sample size of mapping groups are depicted in Fig. 3c. **b** smORFs located within repeat regions. **c** smORF classification based on their genome location. **d** Number of conserved Rp3 smORFs across eukaryotes. For all box plots, the center line corresponds to the median; box limits represent the upper and lower quartiles; whiskers correspond to the 1.5× interquartile range. Source data are provided as a Source Data file.

RPKMs for Ribo-Seq reads mapping to Rp3 microproteins identified with HLA peptidomics seems shifted towards lower values, which suggests these microproteins could have faster turnover. Moreover, we have identified significant evolutionary conservation for most Rp3 microproteins (Fig. 6h), and only 122 microproteins were not found in the analyzed eukaryotic genomes. The vast majority of conserved microproteins are those that also have Ribo-Seq coverage.

## Discussion

The orthogonal approach of using both proteogenomics and Ribo-Seq to identify smORFs we employed in this study is particularly important to perform a truly comprehensive microprotein identification analysis. We showed how limitations in the read alignment step of Ribo-Seq reads prevent the correct annotation of unannotated smORFs that can be found with proteogenomics unambiguously. Sequence-wise, there are two main reasons for this discrepancy: the presence of repeat regions and paralogous sequences for the smORFs in the genome, and multiple different mRNA isoforms that carry the same or a similar smORF.

Gene paralogy results in reads that map to multiple regions in the genome during the alignment step, called multi-mappers, which are often discarded after a certain threshold[17]. Reads that map to a genomic region that encodes many different transcript isoforms are often treated as ambiguous reads, which are disregarded by default by read counting tools such as featureCounts. Simply including all ambiguous and multi-mapping reads into the analysis is not advisable, however, as this would increase the number of false positives and add uncertainty about the coordinates the reads are, in fact, mapping to. Moreover, the number of uniquely assigned reads in our datasets is greatly undermined by the number of multi-mapping reads, which reinforces the idea of not simply including non-uniquely mapped reads. Proteogenomics solves that by allowing us to unambiguously assign a Unique Tryptic Peptide (UTP) to a

microprotein in the custom database. It is worth noting that just because a smORF has Ribo-Seq reads above a certain threshold, it is no guarantee that it would be called by Ribo-Seq-only pipelines, as the tools require, among other features, coverage across the whole smORF sequence, 3-nt periodicity, and a usual build-up of reads around the start codon with a slope around the stop codon. By leveraging proteomics evidence, we are able to circumvent this by identifying peptide evidence in a region that would otherwise be too similar to other parts of the genome, even if the rest of the sequence is not. If the remaining portion of the sequence has lower homology, the Ribo-Seq reads could still map to it without multi-mapping, but the highly conserved portion would confound Ribo-Seq callers. This is the case with the representative smORF in Fig. 5, where a higher degree of homology towards the end of the sequence confounds Ribo-Seq analysis, but still allows the MS peptide to map unambiguously. To add to the reliability of the identifications, most Rp3 microproteins in both the mouse (Fig. 4d) and human (Fig. 6h) datasets were highly conserved in other eukaryotes. A few, however, seem to be species-specific, even with Ribo-Seq coverage. Nonetheless, this is in agreement with findings reporting substantial evidence for the emergence of de novo genes[33–35], especially short ones, reiterating the importance of not disregarding evolutionarily new microproteins.

The annotated regions in the genome overlapped by novel smORFs provide some insights into the role of these novel sequences. In this study, instead of using traditional ways of annotating smORFs, we leveraged information that allowed us to better understand the reason behind the mapping limitations of very short Ribo-Seq reads. A big proportion of smORFs in MM and MM_Amb clusters were classified as rtORFs, meaning they share the loci with either pseudogenes or retrotransposons. This is very informative, as pseudogenes typically arise from the duplication of functional genes[36], and retrotransposons are present in multiple copies across the genome due to their highly

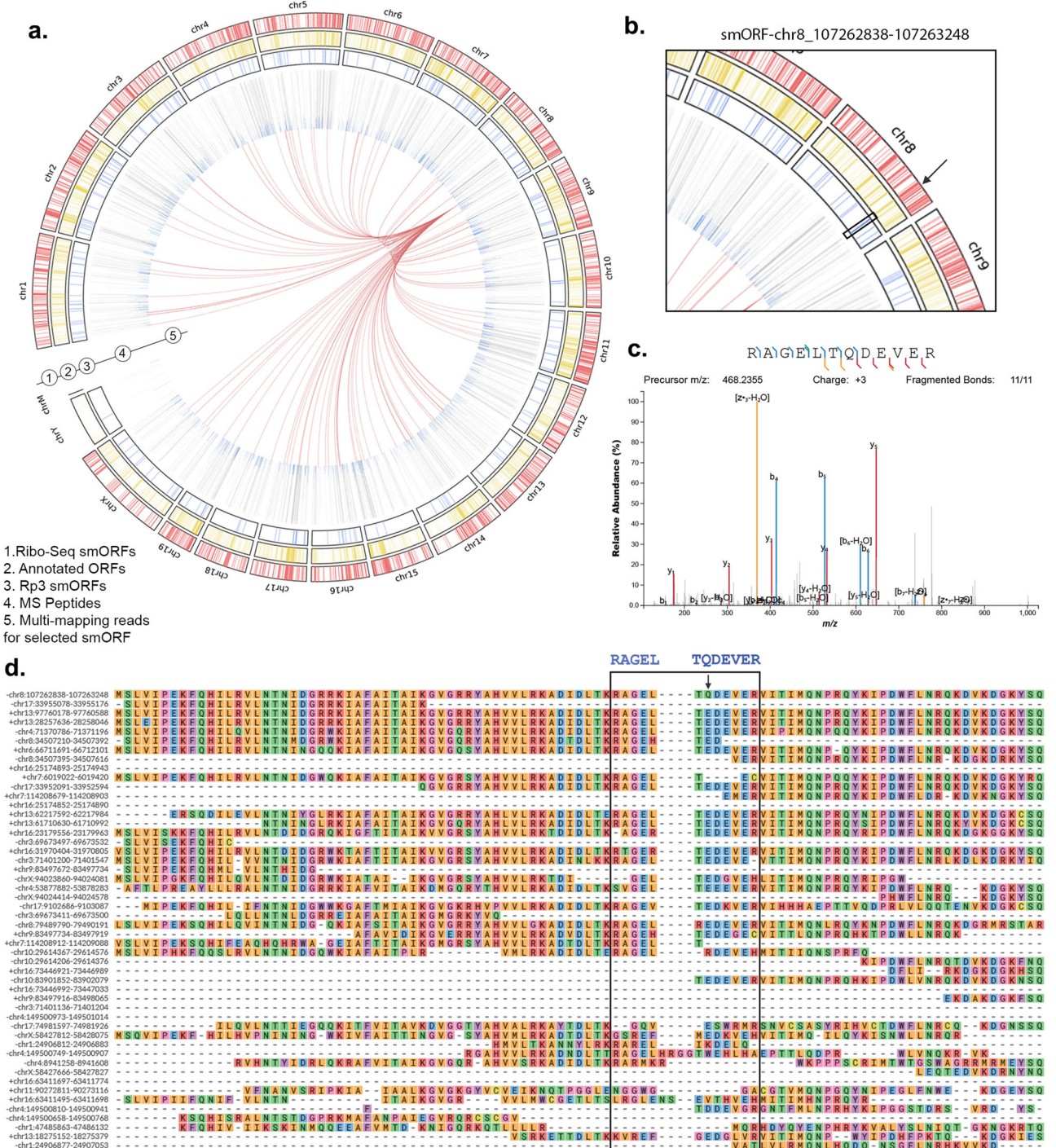

**Fig. 5 | Mapping landscape of the representative rtORF−chr8:107262838−107263248. a** Circos plot illustrating the genomic landscape of unannotated smORFs identified by Ribo-Seq (outermost ring, red bars) and proteogenomics (third ring from border to center, blue bars). The second ring from border to center contains the coordinates of annotated transcripts. The fourth ring contains peaks showing the total number of proteins a microprotein peptide maps to. Blue peaks indicate the presence of a tryptic peptide that does not match any predicted microprotein from the three-frame translation, thus representing truly unique-tryptic peptides (UTPs). The links in the center represent the other sites in the genome that the reads that map to the representative smORF also map to. **b** Zoom-in of the representative smORF in the circos plot to highlight its genome position and features. **c** Annotated fragmentation spectra showing the fragment ions for a peptide of the microprotein encoded by the representative smORF. **d** Multiple sequence alignment (MSA) of the protein sequence of the representative smORF with all its homologs in the mouse genome. The peptide sequence from the fragmentation spectra in **c** is shown in its position in the protein sequence of the smORF in the MSA, highlighted by a black box. A black arrow denotes the amino acid that distinguishes the microprotein UTP from all its homologs.

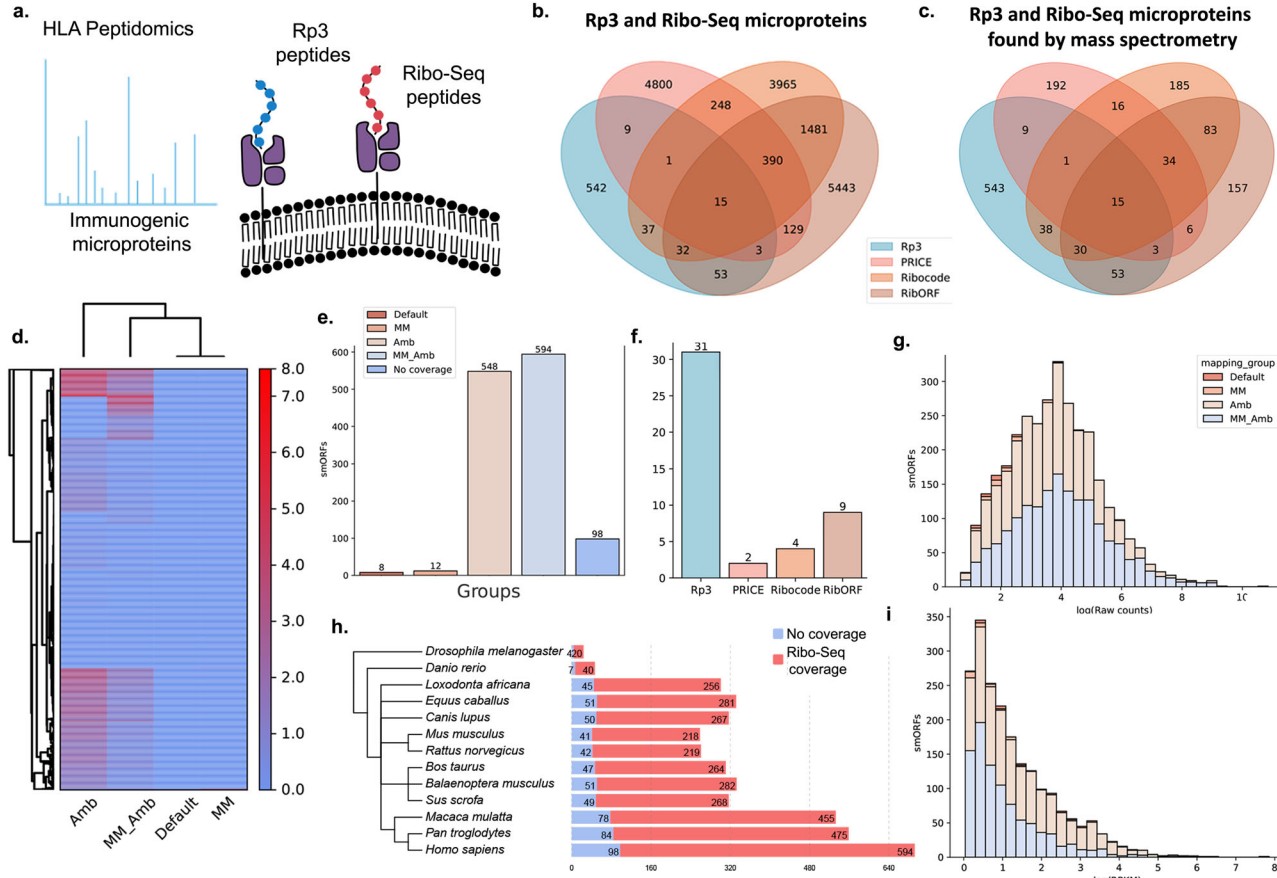

**Fig. 6 | Identification of microproteins in HLA peptidomics datasets.**
**a** Schematic drawing illustrating the presence of proteogenomics and Ribo-Seq microproteins presented on the HLA complex. **b** Venn diagram showing the intersection between Rp3 and Ribo-Seq-only microproteins. **c** Venn diagram showing the intersection between Rp3 and Ribo-Seq-only microproteins with proteomics evidence. **d** Heatmap showing Ribo-Seq coverage in log RPKM for Rp3 smORFs. **e** Number of Rp3 smORFs with Ribo-Seq coverage among the different clusters related to parameters allowed during read counting (same as Fig. 3b). **f** Which of the Rp3 multi-mapping smORFs, including those in MM or MM_Amb groups but absent from Amb, that were also identified using Ribo-Seq-only pipelines. **g** Distribution of raw read counts for Rp3 smORFs. **h** Number of conserved Rp3 smORFs across eukaryotes (**i**) Distribution of RPKM values for Rp3 smORFs.

dynamic copy-paste mechanism[37]. Thus, these two classes of genes are expected to have high sequence similarity to other sites in the genome, resulting in regions prone to multi-mapping. The low numbers of uORFs across all groups is especially informative, as this is the most common type of smORFs to be identified in Ribo-Seq studies. This means that proteogenomics and Ribo-Seq pipelines might be more suitable to identify distinct groups of smORFs each, regarding their classification. As the number of Rp3 smORFs with no Ribo-Seq coverage is low but still exists in both proteomics datasets we analyzed, we advise to prioritize Rp3 microproteins that have at least coverage when allowing ambiguous and/or multi-mapping during read counting to avoid the inclusion of hits that have no translational evidence whatsoever. Rp3 microproteins with Ribo-Seq coverage and pure Ribo-Seq hits from established pipelines with mass spectrometry evidence are thus considered the gold standard for microprotein identification in this study.

Moreover, to reduce the number of false positives, we performed a second round of searches during the proteogenomics analysis by generating a database consisting of annotated proteins plus the microproteins identified in the initial proteogenomics searches, which used the whole 3-frame translated database from the transcriptome. This way, the FDR assessment of proteogenomics hits is not that strongly affected by a bloated database consisting of millions of sequences and we can achieve an accurate global FDR. By reassessing the FDR and checking the Ribo-Seq coverage for Rp3 microproteins, we are left with a very high-confidence set of novel sequences. To control FDR separately for unannotated microproteins identified by Rp3 and canonical proteins, we also employed a grouped-FDR approach, where we separated these two groups of microproteins and assessed the FDR separately with Percolator for each group, for both the mouse (Supplementary Fig. 4) and human (Supplementary Fig. 5) datasets. However, this greatly increased the number of identifications. To be more conservative and avoid including false positives, we kept the results using the approach where we assessed the FDR for the microproteins and canonical proteins together. Hits without Ribo-Seq coverage, although less reliable, are still robust as we included many additional steps in the pipeline to properly control the FDR. In the lack of Ribo-Seq data, Rp3 may still be used like a more traditional proteogenomics pipeline.

An important aspect to note is that the overlap between the Ribo-Seq and proteogenomics results was negligible, as very few microproteins were shared between the approaches. This is surprising, as we expected a higher overlap between microproteins with proteomics evidence and microproteins with translational evidence. This reinforces the complementary aspect of these two different approaches, as each returns a unique subset of microproteins that could not be identified with the other method due to differences in sequence composition. This corroborates findings from a recent study reporting a small intersection of results from Ribo-Seq-only pipelines identifying microproteins encoded by smORFs[38]. Furthermore, the HLA peptidomics analysis provides a reliable source of evidence for novel

microproteins without limiting the search space due to enzymatic cleavage of the proteome, potentially enabling its annotation based on antigen presentation. Also, it shows that this pattern is neither limited to the mouse genome or to the presented Ribo-Seq and proteomics datasets from a specific study[26]. We would like to reiterate that Ribo-Seq-only pipelines are still a very reliable way to identify smORFs, especially due to their high sensitivity. However, using a Ribo-seq-assisted proteogenomics workflow allows the identification of a subset of microproteins residing in regions largely inaccessible to pipelines using translational evidence alone. We expect this workflow to increase proteomics coverage for Ribo-Seq results as well as to improve the quality of microprotein identification pipelines, as those microproteins with Ribo-Seq and proteomics coverage are the most compelling for downstream biological studies.

## Methods

### Reference-guided transcriptome assemblies

To maintain consistency with the previous analysis, we obtained the GTF files produced by Martinez et al. (2023)[26] for brown, beige, and white mouse adipose tissue. These files were generated after trimming the reads, removing the ones that mapped to rRNAs and tRNAs and aligning the remaining reads to the genome with STAR. The transcriptome was assembled using StringTie with default parameters to identify novel transcripts. For human data, we used GTF files from the cell lines HeLa-S3, and K562 from Martinez et al. (2020)[2] and HEK293T from Ma et al. (2018)[39]. These GTF files were generated after aligning trimmed and ribosomal-clean reads to the human genome (assembly hg19) with STAR. The transcriptome was assembled using Cufflinks with default parameters while requiring multi-mapping and fragment bias correction. The reads for these three cell lines were obtained from total RNA-Seq ran on Illumina HiSeq 2500 and NextSeq 500.

### Generation of custom databases for proteomics

To generate a database containing the whole coding potential of each cell line, we translated each transcriptome assembly generated to the three reading frames with the script GTFTToFasta[2], giving priority to ATGs. In the case of multiple ATGs for a region between two stop codons, the most upstream ATG was selected. For the mouse datasets, we generated a non-redundant GTF file containing the merged transcriptomes for the three assemblies. This was done because the mass spectrometry files in the study were generated by combining the tryptic digests of the secretomes and proteomes of all three adipose tissue phenotypes. Afterward, we appended the Swiss-Prot Mouse proteome from Uniprot (UP000000589) to the database and tagged each annotated sequence to be removed in the post-processing steps. For each resulting database, we generated a decoy database containing the reverse sequences of each protein plus the contaminants, which were appended to the initial database. We performed the same bioinformatic steps for the HLA peptidomics proteogenomics analysis but used previously assembled human transcriptomes from HEK293T[39], HeLa-S3, and K562 cell lines[2] to generate the custom databases, to which we appended the Swiss-Prot human reference proteome from Uniprot (UP000005640). We have implemented this step in the *database* mode of the Rp3 pipeline.

### Peptide search and post-processing of mass spectrometry data

To identify microproteins with Rp3, we used MSFragger (v3.5)[40] to map fragmentation spectra to the custom databases generated in the last step. For mouse datasets we used proteomics datasets that were previously generated for mouse brown, beige, and white adipose tissues. We used DDA files from whole-cell lysates and whole-secretome proteomes. The parameters were set as follows: precursor mass tolerance of 20 ppm, fragment mass tolerance of 50 ppm, Data-dependent acquisition as data type, fragment mass tolerance of 0.02 Da, and included carbamidomethylation (+57.021464 Da) as a fixed

modification, and oxidation of methionine (+15.9949 Da) as a variable modification. For the validation step, we used Percolator (v3.06.1)[41] to infer the FDR with a cutoff set at 0.01 at the protein level using the `–picked-protein` method. All pin files from the same datasets were processed together to better infer the FDR. After the search, we removed any peptides that matched an annotated protein but kept those that matched more than one predicted microprotein from the three-frame translated database. To make sure no annotated sequence was included in the results, we performed a Blastp (v2.12.0+) search against NCBI Refseq and excluded any microproteins that had a hit with identity and query cover 100%, i.e., a perfect match. Lastly, we performed a second round of searches to re-score the identified peptides by appending the final set of microproteins from the first search to the same reference proteome. Then, this smaller database was searched using the same mass spectrometry data and following the same bioinformatics steps. This was done to accurately assess the FDR without the effect of bloated databases coming from the three-frame translation of the transcriptome. During re-scoring, we applied MSBooster (v1.2.1)[42] after the peptide search with MSFragger to predict retention times (RT) for each peptide using DIA-NN[43]. The `delta_loess_RT` values were included in the 'pin' file generated during the peptide search as an extra feature for Percolator, so its model would include the calibrated RT values during prediction, which correspond to the experimental RT mapped to the predicted scale of a local regression.

To process HLA peptidomics data, since we are looking for peptides with no specific enzymatic cleavage, such as trypsin's, we specified nonspecific cleavage with the parameters `-- search_enzyme_cutafter ARNDCQEGHILKMFPSTWYV` and `--search_enzyme_name` nonspecific. Additionally, we specified the parameters `--num_enzyme_termini` 0, `--precursor_true_tolerance` 6, `--digest_mass_range` 500.0_1500.0, `--max_fragment_charge` 3, `--digest_min_length` 8, `--digest_max_length` 25. Since the evidence provided by HLA peptidomics is the intact peptide, and a protein does not undergo experimental digestion beforehand, we applied a cutoff for an FDR of 0.01 at the peptide level. Similarly to the mouse datasets, we performed the same re-scoring steps and applied MSBooster to predict RT retention times. After removing every peptide that matched an annotated protein, we ran mhcflurry-predict-scan from the MHCFlurry package to identify which microproteins would contain peptides with strong binding affinity to the MHC Class I using the alleles HLA-A*02:01,HLA-A*03:01,HLA-B*57:01,HLA-B*45:01,HLA-C*02:02,HLA-C*07:02,HLA-A*01:01,HLA-A*02:06,HLA-B*44:02,HLA-B*07:02,HLA-C*01:02,HLA-C*03:01. We selected peptides with an affinity ≤ 100 and an affinity percentile ≤ 0.02. Only peptides 7–12 aa long were included (--peptide-lengths 7–12).

To check for proteomics coverage for the microproteins encoded by the smORFs identified with Ribo-Seq, we used the same parameters for MSFragger and Percolator and applied similar filters. The databases consisted of microproteins identified by either PRICE, Ribocode, or RibORF in each cell line. Since the database was much smaller than the proteogenomics one used for Rp3, we did not re-score the results from the first search to increase the number of identifications and keep fair comparisons. To identify which Rp3 microproteins would be annotated in Uniprot unreviewed databases, we performed a string match for each mass spectrometry peptide and protein in the Uniprot database and classified them based on their annotation level, which indicates how well-characterized the protein is. We have implemented the peptide search and post-processing of mass spectrometry data in the *search* mode of the Rp3 pipeline.

### Processing of Ribo-Seq data

We trimmed the Ribo-Seq reads to remove the Illumina adapters (AGATCGGAAGAGCACACGTCT) using fastx_clipper with the parameters `-Q` 33, `-l` 20, `-n`, `-v`, `-c`, piped to fastx_trimmed with the parameters `-Q` 33, `-f` 1, both tools from the FastX Toolkit (v0.0.13)

(http://hannonlab.cshl.edu/fastx_toolkit/). To remove reads mapping no tRNAs and rRNAs, we mapped the trimmed reads to a contaminant database running STAR with `-outSAMstrandField intronMotif, --outReadsUnmapped Fastx`. To map reads to the genomic coordinates of Rp3 smORFs, we used unmapped reads in fastq format as input for STAR with the parameters `--outSAMstrandField intronMotif, -outFilterMismatchNmax 2, --outFilterMultimapNmax 9999, --chimScoreSeparation 10, --chimScoreMin 20, --chimSegmentMin 15, --outSAMattributes All`. We specified a max of 9999 multi-mappings so STAR would not discard all reads mapping up to that many regions, as it does not keep reads with a number of alignments above the specified threshold. Reads for RibORF, Ribocode, and PRICE were aligned specifying a `--outFilterMultimapNmax 4`, as these tools use Ribo-Seq evidence alone. The Rp3 pipeline is more permissive because it processes smORFs identified with proteomics evidence, and uses the reads only to check for coverage for these microproteins. RibORF was run with default parameters, keeping only the ones with a score ≥0.7. Ribocode and PRICE both require CDS and start and stop codons in the GTF file to generate metaplots and predict 3-nt periodicity. As we are working with reference-guided assemblies from String-Tie and Cufflinks, these GTF files contain only transcript and exon features. To include CDS features, we ran Transdecoder (v5.7.1) (https://github.com/TransDecoder/TransDecoder) to predict the longest ORF in each transcript and then appended the predicted CDS to the GTF files. These predicted CDS are only used as a starting point for predicting 3-nt periodicity, but ORFs are predicted from other putative regions in the transcripts later by the tools regardless. For each transcriptome assembly with CDS information, we generated a STAR (v2.7.4a) index with the custom annotations specified with the parameter `−sjdbGTFfile`. For PRICE, reads were aligned the same way as for RibORF. For Ribocode, since it requires reads to be aligned to the transcriptome and not the genome, we ran STAR with the parameters `−quantMode TranscriptomeSAM GeneCounts −outFilterType BySJout and −alignEndsType EndToEnd`, as suggested in their documentation.

We chose to infer the 3-nt periodicity and generate metaplots using each software's own method, instead of defining the same off-sets for every analysis. This was done to keep consistency with the method of each tool, since they also calculate statistics based on their own predictions. To score ORFs with PRICE, we ran it specifying an FDR of 0.01. Since it does not generate protein sequences or GTF files, we ran $ gedi -e ViewCIT to generate results files filtered with the specified FDR and generated a custom GTF file based on the coordinates, which was used to generate a fasta file containing protein sequences with Gffread[44]. To score the ORFs with Ribocode, we first ran $ prepare_transcripts specifying the adequate custom GTF file for each condition and the same reference genome used when preparing the indexes. Then, we ran $ metaplots and $ Ribocode with default parameters. Afterward, we filtered the results, applied a cutoff for the adjusted *p*-value (FDR) of 1%, and removed any ORF that was annotated by blasting them to the NCBI RefSeq, or that had a sequence longer than 150 amino acids or 450 codons. The same filtering was applied to results coming from PRICE.

### Analysis of ambiguous and multi-mapping reads
To check for Ribo-Seq coverage for the smORFs we could find peptide evidence for with proteogenomics, we first appended the coordinates of those smORFs to the mouse mm18 reference GTF file. Then, we performed four rounds of read counting with featureCounts (v1.6.3)[31], first with default parameters and then allowing multi-mapping reads with `−M`, allowing ambiguous mapping reads with `−O`, and allowing both ambiguous and multi-mapping reads with `−M` and `−O`. We plotted the counts for each combination of these two parameters (-M: MM, -O: Amb, -M -O: MM_Amb, and Default, when using default parameters) into a heatmap using the Python package nheatmap (https://pypi.org/project/nheatmap/). We then classified the smORFs based on their detectability using the different parameters of featureCounts. A

smORF is included in a specific group if its set of parameters results in at least 10 raw counts for that smORF. The mapping classification for proteogenomics smORFs can be obtained by running the Rp3 pipeline in the *ribocov* mode. For each group, we performed comparisons regarding a number of isoforms and repeated regions as follows: first, for the overlapping features, we ran bedtools intersect (v2.30.0) (ref. 31) using the reference GTF file to identify how many features overlapped each smORF, and plotted those into box plots. To investigate repeat regions, we used repeat files for the mm10 genome provided by RepeatMasker (http://www.repeatmasker.org/species/mm.html) and checked which smORFs overlapped the repeat coordinates. For overlapping regions, we classified the overlap with the reference GTF file for mm10 from Ensembl after running bedtools intersect on the reference and custom GTF file containing the smORFs.

### Circos plot generation
To generate the circos plot for read mapping visualization at the genome level, we used the Python package PyCircos (v0.3.0) (https://github.com/ponnhide/pyCircos) and added bars corresponding to the coordinates of each gene annotated in the reference mm10 GTF file to the outermost ring. Then, we did the same for the smORFs found with Ribo-Seq and for the Rp3 smORFs, which were added to the second and third ring, from edge to center, respectively. To analyze the multi-mapping landscape and generate the circos plot, we performed additional filtering in the sam files to analyze the distribution of reads with secondary alignments across the genome. For each sam file, we first kept only the reads with the flag 0x100, which indicates a secondary alignment, by running Samtools (v1.13) with the parameter `-f 0x100`. Afterward, we iterated the alignments and selected the reads that aligned at least once to a region in the genome that encodes an Rp3 microprotein based on the coordinates of our custom GTF file, while keeping all the other regions these selected reads mapped to. Using the filtered sam file containing only reads that mapped to at least one Rp3 smORF, we selected the reads that mapped to the representative smORF. Afterward, we added links to the center of the plot whose coordinates they point to in the genome correspond to the other regions the reads map to. To generate the multiple sequence alignment, we first identified homologs by running tblastn on the mouse genome assembly mm10. Then, we performed the multiple sequence alignment (MSA) with these sequences using MAFFT[45] and generated the MSA figure using the msa4u tool from the package uORF4u[46].

### Sequence analysis comparison among annotated, Ribo-Seq and proteogenomics microproteins
To plot the amino acid composition, we extracted the microproteins sequences for each group of microproteins and plotted the distribution with the Python package matplotlib. To obtain the number of UTPs for each microprotein and the isoelectric point for each peptide, we used the tool RapidPeptidesGenerator (RPG)[47], using trypsin as the enzyme for the in silico protein digestion, and plotted the distributions with matplotlib.

### Conservation analysis
To identify Rp3 smORF homologs in the genome of other eukaryotic organisms and check their overall level of conservation, we a performed a tBlastn search against the genome assemblies of *Equus caballus* (GCF_002863925.1), *Homo sapiens* (GCF_000001405.40), *Canis lupus* (GCF_000002285.5), *Sus scrofa* (GCF_000003025.6), *Bos taurus* (GCF_002263795.2), *Pan troglodytes* (GCF_028858775.1), *Rattus norvegicus* (GCF_015227675.2), *Danio rerio* (GCF_000002035.6), *Balaenoptera musculus* (GCF_009873245.2), *Loxodonta africana* (GCF_000001905.1), *Drosophila melanogaster* (GCF_000001215.4), *Macaca mulatta* (GCF_003339765.1). Then, we filtered the alignments to include only those with an *E*-value < 0.001 and a bitScore > 50. We then annotated a common NCBI tree in phylip format for these species

using EvolView[48] by adding bars to the side of the tree corresponding to the smORFs identified with Rp3 in the organism of interest and numbers of smORF homologs in each species, according to their Ribo-Seq coverage.

## Statistical analysis

We used the Python package SciPy (https://scipy.org/) to perform statistical analysis when adequate. To compare metrics such as the number of peptides per protein (Fig. 2f), molecular weight (Fig. 2g), p*I* (Fig. 2h), and number of overlapping features in the genome (Fig. 4a), we first combined the microprotein results coming from the same tool that passed the FDR thresholds. We performed comparisons based on the combined results of each tool ($n = 5$ biological replicates for a tryptic digest pool for mass spectrometry experiments. $n = 2$ biological replicates for each phenotype for Ribo-Seq experiments). To compare distribution of read assignment (Fig. 3d), we ran featureCounts on each RNA-Seq ($n = 2$ biological replicates for each of the three adipose tissue phenotypes for total RNA and another 2 for mRNA, all grouped together) and Ribo-Seq ($n = 2$ biological replicates for each of the three adipose tissue phenotypes, all grouped together) fastq files and compared the summary output containing the number of assigned reads for each group. We first checked for a Gaussian distribution for each group using the Shapiro–Wilk test. As our data followed a non-parametric distribution (Fig. 2f–h), we ran the Kruskal–Wallis test followed by Dunn's post hoc adjusted for multiple hypothesis testing with the Benjamini-Holchberg FDR correction using SciPy. To perform comparisons between the read assignment of RNA-Seq and Ribo-Seq (Fig. 3d), since the data followed a parametric distribution, we ran *T*-tests using SciPy. We added statistical significance to the plots using statsannotations (https://github.com/trevismd/statannotations).

## Reporting summary

Further information on research design is available in the Nature Portfolio Reporting Summary linked to this article.

## Data availability

Raw data from mouse Ribo-Seq and mass spectrometry experiments used in this study were previously generated by Martinez et al. (2022)[25] and downloaded from Gene Expression Omnibus (GEO) under accession code GSE198109 [https://0-www-ncbi-nlm-nih-gov.brum.beds.ac.uk/geo/query/acc.cgi?acc=GSE198107] and MassIVE under accession codes MSV000089022 [https://massive.ucsd.edu/ProteoSAFe/dataset.jsp?task=220e5ba9df934a34a5eff5dc93081a7c] and MSV000089023 [https://massive.ucsd.edu/ProteoSAFe/dataset.jsp?task=1aef9f40f32b45488349ab0bfc5dfa17], respectively. smORF sequences previously identified by Ribo-Seq and raw Ribo-Seq data were obtained from Martinez et al. (2020)[2] under GEO accession code GSE125218. HLA peptidomics datasets were previously generated by Bassani-Sternberg (2015)[32] and are available at proteomeXchange under accession code PXD000394. Information regarding novel microproteins identified in this study is available in Supplementary Data 1 and 2. GTF files comprising Rp3 identifications are available in Supplementary Data 3. Source data are provided with this paper.

## Code availability

A command-line Python implementation of the Rp3 pipeline under the GNU General Public License (GPLv3) and custom scripts written for this study are available at https://github.com/Eduardo-vsouza/rp3[49], covering the steps for proteogenomics analysis, and Ribo-Seq coverage for proteogenomics microproteins.

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

## Acknowledgements

A.S. acknowledges financial support from NIH grants P30CA014195, R01GM102491, and RC2DK129961; Frederick Paulsen and the Ferring Foundation; and a sponsored research agreement with Novo Nordisk Research Center Seattle, Inc. This work was supported in part by National Institute of Science and Technology on Tuberculosis (Decit/SCTIE/MS-MCT-CNPq-FNDTC-CAPES-FAPERGS) [grant number 421703/2017-2], Banco Nacional de Desenvolvimento Econômico e Social (BNDES/FUN-TEC) [grant number 14.2.0914.1], FAPERGS [grant numbers 17/1265-8 INCT-TB and 19/1724-3 PQG] and CAPES (CAPES-Print program 041/2017). C.V.B. (CNPq, grant 311949/2019-3) is a Research Career Awardee of CNPq. This study was financed in part by the Coordenação de Aperfeiçoamento de Pessoal de Nível Superior—Brasil (CAPES), Finance Code 001. We would like to acknowledge the financial support given by Coordenação de Aperfeiçoamento de Pessoal de Nível Superior—Brasil (CAPES). C.V.B. would like to acknowledge the financial support given by CNPq/FAPERGS/CAPES/BNDES to the National Institute of Science and Technology on Tuberculosis (INCT-TB), Brazil. We would like to thank Manami J. Nishikawa for helping design the schematics for Figs. 3 and 6.

## Author contributions

E.V.S. and A.S. conceptualized the study. E.V.S., C.A.B., B.M. and A.L.B. performed the bioinformatics methodologies, visualization, and data curation. E.V.S. wrote the python code for the bioinformatics pipeline. A.S., C.V.B., P.M., and L.A.B. performed supervision of the research. E.V.S., A.S., and C.V.B. wrote the original draft of the manuscript.

## Competing interests

The authors declare the following competing interests. Angie L. Boookout and Christopher A. Barnes are employees of Novo Nordisk. Alan Saghatelian is a consultant for and cofounder of Exo and Velia Therapeutics. All other authors have no competing interests.
