## [Transparent Peer Review file · Nature Communications]

Rp3: Ribosome profiling-assisted proteogenomics improves coverage and confidence during microprotein discovery

Corresponding Author: Professor Alan Saghatelian

Version 0:

Reviewer comments:

Reviewer #1

(Remarks to the Author)

De Souza et al. presented an approach to better understand the highly discordant rate of identification of microproteins using Ribo-Seq and what they refer to as 'proteogenomic' (RNA-seq based) data. Specifically, they found that repeat intergenic regions and paralogous sequences are the two main causes for the discrepancy. While the manuscript has interesting observations, there are number of issues that are related to both technical (methods) aspects as well as presentation clarity.

1) First, and most important, the author observe that removal of non-unique (multi-mapping) reads in ribo-seq data analysis leads to the loss of identification of microproteins that otherwise can be detected from the RNA-seq data. This represents the central focus of the manuscript. However, I do not agree that simply removing those filters is a solution. Uniqueness filters were introduced to remove false positives in the first place. How can one trust an identification of a new protein if it is not supported by at least some solid evidence? Furthermore, finding peptides in mass spectrometry data for those microprotein found from non-unique reads is not sufficient. Those peptides, as ribo-seq/rna-seq reads, may also map to multiple other coding proteins. For proteomics analysis, the authors used a reference proteome database, but there is no guarantee those peptides would not map to other sequences using extended database containing alternative splice forms etc. At the very least they should have used full UniProt (including TrEMBL sequences) and Ensembl databases.

2) The terminology 'proteogenomic pipeline' vs rna-seq pipeline and ribo-seq pipeline is confusing. Figure 1 is not easy comprehend. Introduction of new terms such as 'proteogenomic proteins' is unnecessary. The figures are misleading. I still do not understand the difference between the Venn diagrams in Fig 2a and 2b.

3) Identification of microproteins using mass spectrometry data requires group-specific FDR approach. This has not been done, so the FDR for the subset of microproteins has not been well controlled.

4) Deep learning-based predictions of retention times an spectra could be used to increase the confidence of identification of peptides from novel microproteins. It's a very useful strategy that the authors unfortunately did not apply.

5) The authors varied the parameters of featureCounts to benchmark the performances of the pipeline in handling multi-mapped reads. However, this would not address the effect of RNA alignment step on multi-mapping. The author should also try another RNA aligner, such as bbmap, at least on a subset of data and report the results. In Figure 1, the authors mentioned HISAT2, but didn't mentioned it in the Methods and the rest of the manuscript.

6) It appears that access <https://github.com/Eduardo-vsouza/rp3> is not allowed. It would be important in order to have a better understanding of the pipeline. The authors mentioned in Figure 1 that they performed RNA assembly with Stringtie, but there is no follow up in Methods or the rest of the manuscript. What are some statistics of the assembled transcriptome?

7) What is the overall QC (coverage, number of reads) of the Ribo-seq samples? If these libraries were not sequenced deep enough, that could explain the difficult to detect the coverage of some smORF.

8) How conserved are the genes that generated eORF?

Reviewer #2

(Remarks to the Author)

I co-reviewed this manuscript with one of the reviewers who provided the listed reports. This is part of the Nature

Communications initiative to facilitate training in peer review and to provide appropriate recognition for Early Career Researchers who co-review manuscripts.

Reviewer #3

(Remarks to the Author)

The paper "The Integration of Proteogenomics and Ribosome Profiling Circumvents Key Limitations to Increase the Coverage and Confidence of Novel Microproteins" presents a combined ribosome profiling and proteogenomics pipeline for the identification of small translated ORFs. They argue that multi-mapping of reads is a significant problem for ribosome profiling and that combining this data with data from proteogenomics can alleviate this problem. While it is not novel to combine proteomics and ribosome profiling, the specific pipeline is new.

The study points out that some smORFs are not detected by ribosome profiling, but can be found by proteogenomics. They mostly attribute this to ribosome profiling reads being short and therefore prone to not map uniquely, but instead map to many loci.

A somewhat weak aspect of this study is that it is not necessarily addressing a shortcoming of ribosome profiling as such, but more highlights the weaknesses of a specific pipeline developed previously by the authors. While the text uses wording such as "current Ribo-seq pipelines" and "disregarded in most pipelines" it appears to only compare to the author's pipeline(s) that is based on the tool RibORF.

The study demonstrates that saving multi-mapping reads is beneficial for ORF detection based on the results presenting depletion of these in peptide-predicted ORFs. It is, however, not clear that all or most "current" pipelines discard all multi-mappers, nor that the standard STAR mapping of reads combined with a different prediction tool than what the authors used will not necessarily capture the missing MS-predicted ORFs. Indeed many of the ORFs are rescued by including multi-mappers.

Obtaining peptide evidence from smORF is obviously solid orthogonal evidence, but to show that proteogenomics can in fact detect many smORFs that ribosome profiling can not, one would have to show that this is still true even if one gives the ribosome profiling tools a fair chance. This could be done by allowing multi-mapping reads to be included when predicting smORFs and using more than just RibORF (which can not make the claim to be a standard in the field). As it is presented there's no reason to believe that these smORFs would not be detected by a less conservative ribosome profiling-based detection pipeline.

What threshold of multi-mapping would one need in order for STAR to map reads to the "missing" smORFs. Are these reads excluded due to the author's ribosome profiling pipeline (e.g. using a cutoff of <4 secondary alignments as stated later)? On the flip side, is there a reason not to include multi-mappers up to 10 as default in STAR, e.g. false positives?

Related to that above. For smORFs that only have coverage from reads that map to very many loci, e.g. >10. These are presumably mostly the rORFs, while the rest uORFs/dORFs etc, are more likely to be in the STAR standard range 2-10 multi-mappers. If the latter is detectable by ribo-seq by using less restrictive filtering, the value of the RP3 approach would be mostly in detecting translated repeats and retrotransposons and perhaps to disambiguate paralogs. To see if this is the case an overview of what types of smORFs that are detectable at what level of multi-mapping should be produced.

Minor considerations

1) Fig 2a - The lack of overlap here is quite concerning. Even accounting for sequence bias and multi-mapping this could not be more than what is expected by random chance.

2) Fig 3B - While the heat map is nice it can easily be misleading due to colouring. It would be therefore be good with plots showing the empirical cumulative distribution of reads over these smORFs. I.e. how many have 0 reads, 1 read, 2 reads ... and so on. This should be both at the read count level and RPKM.

3) "The reads for this smORF mapped to more than 15 different regions in the genome and were filtered by the default parameters of STAR, which has a cutoff of <10 secondary alignments, and by Ribo-Seq pipelines that use a cutoff of < 4 secondary alignments. These cutoffs are necessary for confident mapping due to shorter read lengths. "

As it is stated I don't believe this is true. Several ribo-seq pipelines simply use the standard mapping defaults, e.g. 10 for STAR. If the authors are specifically referring to their pipeline they should write so and if it's a general statement they should at least provide a reference to support this claim.

4) "Most of the time, multi-mapped reads are discarded during read assignment¹⁶, which limits the number of novel smORFs that can be found to be actively translated, especially when working with a genome with a high number of paralogous sequences. "

Reference "16" is Deschamps-Francoeur et al 2020 which is a paper about multi-mapping in RNA-seq. I'm not sure this is always true for ribosome profiling. If you want to make this point about ribosome profiling an appropriate source where they have looked at relevant pipelines should be cited.

Author Rebuttal letter:

Reviewer #1 (Remarks to the Author):

De Souza et al. presented an approach to better understand the highly discordant rate of identification of microproteins using Ribo-Seq and what they refer to as "proteogenomic" (RNA-seq based) data. Specifically, they found that repeat intergenic regions and paralogous sequences are the two main causes for the discrepancy. While the manuscript has interesting observations, there are number of issues that are related to both technical (methods) aspects as well as presentation clarity.

1) First, and most important, the author observe that removal of non-unique (multi-mapping) reads in ribo-seq data analysis leads to the loss of identification of microproteins that otherwise can be detected from the RNA-seq data. This represents the central focus of the manuscript. However, I do not agree that simply removing those filters is a solution. Uniqueness filters were introduced to remove false positives in the first place. How can one trust an identification of a new protein if it is not supported by at least some solid evidence? Furthermore, finding peptides in mass spectrometry data for those microprotein found from non-unique reads is not sufficient. Those peptides, as ribo-seq/rna-seq reads, may also map to multiple other coding proteins. For proteomics analysis, the authors used a reference proteome database, but there is no guarantee those peptides would not map to other sequences using extended database containing alternative splice forms etc. At the very least they should have used full UniProt (including TrEMBL sequences) and Ensembl databases.

We appreciate the feedback and would like to clarify some points that might be unclear in the text. The central idea of the manuscript is to combine the strengths of proteogenomics, RNA-Seq and Ribo-Seq to identify a subset of microproteins that is missed when using the techniques alone. When we remove filters for multi-mapping reads, we do not do so for the traditional Ribo-Seq pipelines that make use of Ribo-Seq reads solely. Instead, we introduce this more permissive approach only for those predicted smORFs that also have peptide evidence from the mass spectrometry data. We agree with the fact that simply removing those filters is not the solution, which is why we kept them in the Ribo-Seq-only pipelines. However, we remove this filter to show that our proteogenomics method is able to provide unambiguous evidence for smORFs that would otherwise have untrustworthy evidence if using translational evidence alone.

We used the reviewed SwissProt database as a proteome as it is the most widely use one for traditional proteomics analysis. We are increasingly seeing in our analysis that indeed, some of the peptides coming from novel microproteins match some proteins in the extended, unreviewed Uniprot database. Those microproteins, however, have very low annotation scores (1-2) and are very poorly characterized. We included an analysis in the paper to show that the peptides that also match to these proteins in the unreviewed Uniprot, are actually matching proteins with low annotation score, so we are providing extra empirical evidence for their existence. Even if our peptides match some of them, we providing more reliable evidence for their existence than the current annotation.

2) The terminology "proteogenomic pipeline" vs rna-seq pipeline and ribo-seq pipeline is confusing. Figure 1 is not easy comprehend. Introduction of new terms such as "proteogenomic proteins" is unnecessary. The figures are misleading. I still do not understand the difference between the Venn diagrams in Fig 2a and 2b.

We appreciate the comments and have changed the terms "proteogenomics pipeline" and "proteogenomics microproteins" to simply "Rp3 microproteins" to make it clear they are coming from the pipeline developed for this study. We also made some changes to figure 1 to improve readability. Regarding the Venn diagrams, we have added captions to the top of the plots in the figure and added some sentences to the text to make it easier to understand their difference. In summary, the first Venn diagram is supposed to show the overlap between Rp3 and the results coming from the (now) 3 Ribo-Seq pipelines. In this diagram, the Ribo-Seq pipelines use translational evidence (Ribo-Seq reads) only. In the second Venn diagram, we show the intersection between Rp3 results and the Ribo-Seq results that we could identified peptide evidence for in the mass spectrometry data. The second Venn is intended to compare the proteomics coverage of the different approaches.

3) Identification of microproteins using mass spectrometry data requires group-specific FDR approach. This has not been done, so the FDR for the subset of microproteins has not been well controlled.

Following the reviewer's suggestion, we assessed the FDR (1%) at the protein level for the mouse datasets using Percolator's "picked-protein" method. For the HLA peptidomics analysis, however, we kept the peptide level FDR, as the experimental design results in peptides presented at the HLA complex. These peptides are not processed by a protease experimentally, and thus represent the peptide evidence for these microproteins. The authors of the original article also did not assess the FDR at the protein level for these datasets. We changed the text to better reflect and describe the methodology choices for the proteomics analysis.

4) Deep learning-based predictions of retention times an spectra could be used to increase the

confidence of identification of peptides from novel microproteins. It's a very useful strategy that the authors unfortunately did not apply.

To increase the confidence of our proteomics analysis, we have incorporated MSBooster into the pipeline to predict retention times (RT) and calculate the ΔRT_{loess} , which represents the calibrated RT values after mapping the experimental RT to the predicted values of their local regression. This feature is incorporated into Percolator's input file to assess the FDR with more accuracy. We made the use of MSBooster to assess the FDR using RT available for users of our pipeline with the tag `MSBooster` during the peptide search mode.

5) The authors varied the parameters of `featureCounts` to benchmark the performances of the pipeline in handling multi-mapped reads. However, this would not address the effect of RNA alignment step on multi-mapping. The author should also try another RNA aligner, such as `bmap`, at least on a subset of data and report the results. In Figure 1, the authors mentioned HISAT2, but didn't mention it in the Methods and the rest of the manuscript.

We appreciate the suggestion but we don't consider the RNA-Seq alignment step to be a solution for this problem. Since both the proteogenomics approach of Rp3 and the Ribo-Seq-only approaches of RibORF, Ribocode and PRICE depend on the transcriptome assembled from the RNA-Seq data, allowing a more permissive or more restrictive alignment of RNA-Seq reads would simply result in a different start point for the overall analysis, but would still be the same starting transcriptome for both the proteogenomics and Ribo-Seq pipelines. It would then change the results of both approaches at the same time, but would not explain why the pipelines tend to identify a distinct subset of proteins.

6) It appears that access <https://github.com/Eduardo-vsouza/rp3> is not allowed. It would be important in order to have a better understanding of the pipeline. The authors mentioned in Figure 1 that they performed RNA assembly with Stringtie, but there is no follow up in Methods or the rest of the manuscript. What are some statistics of the assembled transcriptome?

We tried to include a .zip file containing every file necessary to run the pipeline during the submission process of the manuscript while keeping the GitHub page private until the article is published. Something must have gone wrong, so we have made the GitHub page of the Rp3 pipeline public so the reviewers can check the functionality of the code.

The transcriptomes used in the paper were previously assembled and validated by the studies of Ma et al. (2018) (Ref. 1), Martinez et al. (2020) (Ref. 2), and Martinez et al. (2023) (Ref. 3). We included that in the picture because the assembly is a crucial, initial step to generate the three-frame translated databases for the both the proteogenomics search with Rp3 and the ORF scoring with RibORF, PRICE, and Ribocode. We have added an entire new section in the methodology to explain why the transcriptomes were selected and how they were assembled.

1. Ma, J., Saghatelian, A. & Shokhirev, M. N. The influence of transcript assembly on the proteogenomics discovery of microproteins. *PLoS One* 13, e0194518 (2018).

2. Martinez, T. F. et al. Accurate annotation of human protein-coding small open reading frames. *Nat. Chem. Biol.* 16, 458-468 (2020).

3. Martinez, T. F. et al. Profiling mouse brown and white adipocytes to identify metabolically relevant small ORFs and functional microproteins. *Cell Metab.* 35, 166-183 (2023).

7) What is the overall QC (coverage, number of reads) of the Ribo-seq samples? If these libraries were not sequenced deep enough, that could explain the difficulty to detect the coverage of some smORF. According to our fastQC analysis, the total number of sequences for the 6 mice Ribo-Seq datasets are 82462820, 36729330, 27279700, 27934436, 88556018, 54637215. The Total number of bases are 2.4 Gbp, 1 Gbp, 751 Mbp, 786.6 Mbp, 2.4 Gbp, and 1.5 Gbp. This coverage should be sufficient for most Ribo-Seq analysis.

However, we do not think that a low coverage or number of reads for Ribo-Seq would answer the question about the differences in smORF detection, because we are using the same Ribo-Seq reads to identify Ribo-Seq coverage for the proteogenomics smORFs and to score smORFs with Ribo-Seq-only pipelines. Thus, if the sequencing was not deep enough, it should affect the coverage for the proteogenomics results too.

8) How conserved are the genes that generated eORF?

After applying the protein level FDR and including MSBooster to predict retention times, we could not identify eORFs anymore. The ORF classes now displayed on the paper, in figures 2, 4, and 6, are rORFs, lncRNAs, uORFs, aiORFs, ndORFs, riORFs, dORFs, and non-coding. We provided an explanation for these in the text.

Reviewer #2 (Remarks to the Author):

Reviewer #3 (Remarks to the Author):

The paper "The Integration of Proteogenomics and Ribosome Profiling Circumvents Key Limitations to Increase the Coverage and Confidence of Novel Microproteins" presents a combined ribosome profiling and proteogenomics pipeline for the identification of small translated ORFs. They argue that multi-mapping of reads is a significant problem for ribosome profiling and that combining this data with data from proteogenomics can alleviate this problem. While it is not novel to combine proteomics and ribosome profiling, the specific pipeline is new.

The study points out that some smORFs are not detected by ribosome profiling, but can be found by proteogenomics. They mostly attribute this to ribosome profiling reads being short and therefore prone to not map uniquely, but instead map to many loci.

A somewhat weak aspect of this study is that it is not necessarily addressing a shortcoming of ribosome profiling as such, but more highlights the weaknesses of a specific pipeline developed previously by the authors. While the text uses wording such as "current Ribo-seq pipelines" and "disregarded in most pipelines" it appears to only compare to the author's pipeline(s) that is based on the tool RibORF.

The study demonstrates that saving multi-mapping reads is beneficial for ORF detection based on the results presenting depletion of these in peptide-predicted ORFs. It is, however, not clear that all or most "current" pipelines discard all multi-mappers, nor that the standard STAR mapping of reads combined with a different prediction tool than what the authors used will not necessarily capture the missing MS-predicted ORFs. Indeed many of the ORFs are rescued by including multi-mappers.

Obtaining peptide evidence from smORF is obviously solid orthogonal evidence, but to show that proteogenomics can in fact detect many smORFs that ribosome profiling can not, one would have to show that this is still true even if one gives the ribosome profiling tools a fair chance. This could be done by allowing multi-mapping reads to be included when predicting smORFs and using more than just RibORF (which can not make the claim to be a standard in the field). As it is presented there's no reason to believe that these smORFs would not be detected by a less conservative ribosome profiling-based detection pipeline.

We appreciate the suggestions and have addressed the concerns above by incorporating two other Ribo-Seq tools into the comparison. Figures 2 and 6, for both the mouse and human datasets, now present Venn diagrams showing the overlap between Rp3, PRICE, Ribocode, and RibORF. We tried to keep consistency as much as possible while meeting the alignment requirements for each software. For instance, in the Ribocode pipeline, they suggest a cutoff for multi-mappers of 1 during the alignment step with STAR. PRICE does not specify a suggested multi-mapping cutoff, but employs its own method for rescuing multi-mapping reads. Still, it could not identify most of the microproteins identified with Rp3, although it could identify a unique subset, suggesting an orthogonal approach. To keep consistency with RibORF, we made the alignment step with Ribocode more permissive and allowed the 4 multi-mappings. The same was applied to the alignment preceding scoring with PRICE. We did try a more permissive cutoff for Ribocode and PRICE at 10 multi-mappers, but for Ribocode it did not change the results, and every microprotein was still identified, with no additional ones. For PRICE, the results varied and we lost a subset of microproteins and gained another one. This was probably due to PRICE method of rescuing multi-mapping reads, which would heavily increase the number of false positives by being more permissive. The overall overlap did not seem to change considerably.

What threshold of multi-mapping would one need in order for STAR to map reads to the "missing" smORFs. Are these reads excluded due to the author's ribosome profiling pipeline (e.g. using a cutoff of <4 secondary alignments as stated later)? On the flip side, is there a reason not to include multi-mappers up to 10 as default in STAR, e.g. false positives?

As we stated in the last answer, the suggested cutoff for Ribocode, for instance, is even lower than the 4 multi-mappings. As we are working with reads that are shorter than normal, we made the multi-mapping cutoff more strict than the default 10, as STAR was designed with RNA-seq reads in mind, which are longer. As Ribo-Seq reads are shorter and thus more likely to multi-map by logic, conceptually it makes sense to use a more stringent approach in order to avoid false positives, as the pipelines already identify thousands of novel genes. The statement that Ribo-Seq reads are more likely to multi-map than RNA-Seq reads is now supported by our data in Fig. 3d.

In the case of multi-mappers, yes, by allowing a very high number of multi-mappers, we expect at least some of the Ribo-Seq pipelines to be able to identify the Rp3 microproteins. This would not be viable, however, as just making the cutoff more permissive would introduce possibly thousands of false positives into the analysis. We only suggest the more flexible filters for Rp3 because it already presents the peptide evidence, which solves the ambiguity in ORF detection. We are not trying to make the claim that this should be applied to Ribo-Seq-only pipelines, however. We added some descriptions in the text to make that clearer and keep readers from misunderstanding the proposed approach.

Related to that above. For smORFs that only have coverage from reads that map to very many loci, e.g. >10. These are presumably mostly the rORFs, while the rest uORFs/dORFs etc, are more likely to be in the STAR standard range 2-10 multi-mappers. If the latter is detectable by ribo-seq by using less restrictive filtering, the value of the RP3 approach would be mostly in detecting translated repeats and retrotransposons and perhaps to disambiguate paralogs. To see if this is the case an overview of what types of smORFs that are detectable at what level of multi-mapping should be produced.

We would also like to make a point that, simply by incorporating reads with high number of multi-mappings in the analysis and allowing the Ribo-Seq-only smORFs to have higher coverage would not guarantee their identification by Ribo-Seq pipelines. Differently than Rp3, the Ribo-Seq-only pipelines do not provide peptide evidence that could support and solve the ambiguity problem. Instead, since they rely solely on Ribo-Seq evidence, they need a combination of good coverage across the ORF, clear 3-nt periodicity, a build up of reads around the start codon and towards the stop codon. Simply making the multi-mapping less stringent does not guarantee their identification, as other characteristics such as presence of isoforms could also confound the traditional way of analyzing Ribo-Seq data. For instance, in the case of more than one ORF in the same genome locus but in different reading frames, this would confound Ribo-Seq callers, which highlights the important aspect of overlapping features as well. We reinforce that we do not suggest that Ribo-Seq is not adequate for identifying microproteins, and Rp3 is simply an orthogonal and complementary method to identify a smaller, but unique subset of microproteins. By allowing more flexible filters, we intended to show that the Ribo-Seq coverage is there, but shouldn't be used alone to identify the microprotein. Instead, it could be integrated with proteogenomics to solve the ambiguity issue of Ribo-Seq, while Ribo-Seq solves partially the accuracy and precision problems of pure proteogenomics approaches.

Minor considerations

1) Fig 2a - The lack of overlap here is quite concerning. Even accounting for sequence bias and multi-mapping this could not be more than what is expected by random chance.

The lack of overlap among different approaches to identify microproteins has always intrigued us, and is a problem that has been discussed in the field extensively. This is not unique to Rp3, however, and other study comparing other Ribo-Seq-only tools not tested in this work has also found a very small overlap among the different pipelines (Tong et al., 2023). In our human data, however, we are able to detect a higher overlap between the techniques when compared to the mouse data.

Tong G, Hah N, Martinez TF. Comparison of software packages for detecting unannotated translated small open reading frames by Ribo-seq. bioRxiv. 2023 Dec 30.

2) Fig 3B - While the heat map is nice it can easily be misleading due to colouring. It would be therefore be good with plots showing the empirical cumulative distribution of reads over these smORFs. I.e. how many have 0 reads, 1 read, 2 reads and so on. This should be both at the read count level and RPKM. We have added two plots showing the distribution of both raw read counts and RPKMs for both the mouse and the human data, in figures 4 and 6, respectively. We hope this makes the mapping classification clearer and the heatmap not misleading.

3) The reads for this smORF mapped to more than 15 different regions in the genome and were filtered by the default parameters of STAR, which has a cutoff of <10 secondary alignments, and by Ribo-Seq pipelines that use a cutoff of < 4 secondary alignments. These cutoffs are necessary for confident mapping due to shorter read lengths.

As it is stated I don't believe this is true. Several ribo-seq pipelines simply use the standard mapping defaults, e.g. 10 for STAR. If the authors are specifically referring to their pipeline they should write so and if it's a general statement they should at least provide a reference to support this claim.

We have added specifications for multi-mapping filtering suggested by the developers of Ribocode in their documentation and changed the sentence to better reflect the analysis performed. Additionally, we replaced the representative smORF in Figure 4 (now figure 5, with the introduction of an extra figure). The new microprotein has 50 homologs in the mouse genome and should be a better example to highlight the main point of the paper.

4) Most of the time, multi-mapped reads are discarded during read assignment¹⁶, which limits the number of novel smORFs that can be found to be actively translated, especially when working with a genome with a high number of paralogous sequences. "â"

Reference "16" is Deschamps-Francoeur et al 2020 which is a paper about multi-mapping in RNA-seq. I'm not sure this is always true for ribosome profiling. If you want to make this point about ribosome profiling an appropriate source where they have looked at relevant pipelines should be cited.

To make this statement, we decided to keep the Deschamps reference to add a starting point and introduce the multi-mapping problem but changed the sentence for accuracy. The paper indeed only refers to RNA-Seq, however, so we added an additional analysis showing the differences in read assignment for RNA-Seq and Ribo-Seq reads (Fig. 3d) based on the datasets that were used for our analysis. The box plots show that the distribution of normalized reads differs drastically between unassigned RNA-Seq and Ribo-Seq reads due to multi-mapping, which should confound ORF discovery.

We would like to thank the reviewers for their feedback, as they have greatly improved the manuscript. Our results are much more reliable now and the comparisons much more comprehensive. We believe Rp3 is in a much better shape to help other scientists study microproteins using multi-omics datasets.

Version 1:

Reviewer comments:

Reviewer #1

(Remarks to the Author)

The authors addressed, or at least attempted to address, most of my concerns except this one:

3) Identification of microproteins using mass spectrometry data requires group-specific FDR approach. This has not been done, so the FDR for the subset of microproteins has not been well controlled.

Their response "Following the reviewer's suggestion, we assessed the FDR (1%) at the protein level for the mouse datasets using Percolator's -picked-protein method. For the HLA peptidomics analysis, however, we kept the peptide level FDR, as the experimental design results in peptides presented at the HLA complex..."

indicates that they did not understand my comment about the group FDR. The comment is about the need to separate all peptide-spectrum matches into two (or more) subgroups, e.g. canonical (mapping to UniProt) and noncanonical peptides, and find the 1% FDR threshold for each group separately. Without it, the FDR for Noncanonical peptides, due to their small number, is likely to be significantly underestimated. At the very least they should mention it in the discussion.

Reviewer #2

(Remarks to the Author)

Reviewer #3

(Remarks to the Author)

All my concerns have been addressed.

Author Rebuttal letter:

REVIEWERS' COMMENTS (Author's answers in red)

Reviewer #1 (Remarks to the Author):

The authors addressed, or at least attempted to address, most of my concerns except this one:

3) Identification of microproteins using mass spectrometry data requires group-specific FDR approach.

This has not been done, so the FDR for the subset of microproteins has not been well controlled.

Their response "Following the reviewer's suggestion, we assessed the FDR (1%) at the protein level for the mouse datasets using Percolator's picked-protein method. For the HLA peptidomics analysis, however, we kept the peptide level FDR, as the experimental design results in peptides presented at the HLA complex..."

indicates that they did not understand my comment about the group FDR. The comment is about the need to separate all peptide-spectrum matches into two (or more) subgroups, e.g. canonical (mapping to UniProt) and noncanonical peptides, and find the 1% FDR threshold for each group separately. Without it, the FDR for Noncanonical peptides, due to their small number, is likely to be significantly underestimated. At the very least they should mention it in the discussion.

We performed additional analyses to assess the FDR separately for unannotated microproteins and canonical proteins. We did this by separating the pin file from MSFragger into two different files, each containing either microproteins or canonical proteins, and then ran Percolator once for each group, in order to assess the FDR separately for each. This analysis is mentioned in the discussion and the results are shown in Supplementary Figures 4 and 5.

Reviewer #2 (Remarks to the Author):

Reviewer #3 (Remarks to the Author):

All my concerns have been addressed.

We would like to thank the reviewers for their comments and suggestions. They greatly improved the manuscript and our pipeline.
